# Humoral Immune Response to a Timely Booster mRNA Vaccination in Non-Responders to a Standard Vaccination Schedule against COVID-19 in Kidney Transplant Recipients

**DOI:** 10.3390/jcm11216439

**Published:** 2022-10-30

**Authors:** Julia Stigler, Lukas Buchwinkler, Claire Anne Solagna, Michael Rudnicki, Markus Pirklbauer, Gert Mayer, Julia Kerschbaum

**Affiliations:** Department of Internal Medicine IV, Nephrology and Hypertension, Medical University Innsbruck, Anichstrasse 35, 6020 Innsbruck, Austria

**Keywords:** kidney transplant recipients, COVID-19, mRNA vaccines, non-responders, timely booster vaccine

## Abstract

Kidney transplant recipients who are at increased risk for COVID-19 infection and associated morbidity and mortality have been shown to be prone to an impaired humoral immune response to a standard vaccination schedule against COVID-19 with two doses of SARS-CoV-2 mRNA vaccines. In this study, response rate of 94 kidney transplant recipients without detectable seroconversion after two doses of a mRNA vaccine who were offered a timely third mRNA vaccine after completion of the standard vaccination schedule was retrospectively analyzed. After a median of 28 days, antibody titers against the S1 spike protein showed a non-response rate of 53%. No significant risk factors for non-response could be identified. The responders showed a high variation in antibody titers (median 73.9 BAU/mL, IQR 221.5). In conclusion, a third booster mRNA vaccine in non-responding kidney transplant recipients leads to a detectable humoral immune response in approximately half of the patients. In the seroconversion group, antibody titers were highly variable, indicating that even non-responders to the standard vaccination schedule might develop a significant humoral immune response after a timely booster vaccine.

## 1. Background

Kidney transplant recipients are at an increased risk for severe disease and death from COVID-19 [1] and, hence, were prioritized for vaccination [2] in the beginning of 2021. Subsequent studies evaluating the immune response following a two-dose mRNA vaccination schedule showed diminished humoral or cellular response [3,4,5,6,7,8,9,10,11,12,13], ranging from 36–54%. Similar results were recently published by our group: non-response after the standard vaccination schedule with two vaccines was found in 50% of kidney transplant recipients (*n* = 216) [14]. Parameters which have been shown to be associated with reduced immune response are the use of antimetabolites [15], belatacept, glucocorticoids, and ATG for immunosuppression [5,6,11]; higher age; the type of mRNA vaccine [15]; lower eGFR (estimated glomerular filtration rate) [7,12,16]; lower Hb-levels; and prevalent diabetes [6].

In light of the upcoming evidence of reduced response to the standard vaccination schedule in immunocompromised individuals, the Austrian National Advisory Committee on immunization practices (version 4.2, 5 July 2021) recommended the offer of a third mRNA vaccine to immunocompromised individuals without detectable antibodies after the first two vaccines as an off-label use. A general recommendation for a booster vaccine (dose 3) was released in Austria in August 2021, advising healthy individuals to receive a booster dose 9–12 months after dose 2, and older and/or immunocompromised individuals to receive a booster dose 6–9 months after dose 2.

In some countries, this recommendation of a three vaccination strategy was published earlier. Two studies showed that the three-dose COVID-19 mRNA regimen (days 0, 21, and 49 with Pfizer-BioNTech (Comirnaty, BioNTech/Pfizer, Mainz, Germany)) was safe and the third dose did not increase the risk of local or systemic adverse reactions in kidney transplant recipients [17,18]. A significant humoral immune response (development of neutralizing anti-SARS-CoV-2 antibodies) in initial non-responders was detected in 32.3–45.25% of single organ transplant recipients after the third vaccination. It was demonstrated that nearly all patients, who had low positive antibody titers after the second dose, showed an increase in antibody levels after the third dose [19]. In addition, in a study by Massa et al., T-cell responses to SARS-CoV-2 increased after the third vaccine [17,18].

Nevertheless, the seroconversion rate remained especially low in patients treated with belatacept and/or mycophenolic acid; in patients receiving triple immunosuppression; and in patients who recently received transplantation with high maintenance immunosuppression [20,21,22]. Other risk factors for non-response were lymphopenia, age, and low eGFR [17,18]. In a double-blind placebo-controlled trial, organ transplant recipients received either a third dose of mRNA-1273 or a saline placebo two months after the second dose. While the treatment group had a 60% rate of neutralizing antibody positivity, only 25% had neutralizing antibodies after the third dose in the placebo group. Accordingly, SARS-CoV-2 specific T-cell counts were higher after the third dose in the mRNA-1273 group than in the placebo group [23].

In view of the high risk for COVID-19 infection and associated death in kidney transplant recipients, non-responders to a standard vaccination schedule treated in our center were offered a third mRNA vaccine approximately 3–4 months after the second vaccine. In this study, we retrospectively analyzed the rate of response to the third dose of a mRNA vaccine in initial non-responders.

## 2. Materials and Methods

In this retrospective analysis, the data of kidney transplant recipients who showed no humoral immune response after standard vaccination with two doses of a SARS-CoV-2 mRNA vaccine (BNT162b2 or mRNA-1273) were included. The data were collected from electronic patient records of the Medical University Hospital Innsbruck. This analysis was approved by the Institutional Review Board of the Medical University of Innsbruck (ECS 1015/2022).

According to the prioritization guidelines, the included patients received their initial vaccine doses as early as possible (Q1 2021). During routine clinical practices, antibody response to the vaccination was tested using the “Abbot SARS-CoV-2 IgG II Quant Assay” at 60 to 120 days after the second vaccine. This test is a chemiluminescent microparticle immunoassay used for the qualitative and quantitative determinations of IgG antibodies against the spike receptor-binding domain (RBD) of SARS-CoV-2 in human serum and plasma. Patients, who did not develop a positive antibody titer (defined as a titer above the manufacturer’s specified threshold), were offered an early third mRNA vaccine (median time between the 2nd and the 3rd vaccination was 112 days and at least 4 weeks apart) according to the recommendations of the Austrian National Advisory Committee on immunization practices, version 4.2 (5 July 2021). At this time, this was an off-label use in line with the recommendation for immunocompromised individuals and was administered with informed consent. Four weeks after the third vaccination, antibody titers were measured. The primary outcome was serological response to the vaccine. The limit for non-response was <7 BAU/mL, as defined by the manufacturer. Patients with a history of COVID-19 were excluded, as were patients with present SARS-CoV-2 nucleocapsid antibodies.

Descriptive statistics as well as univariate and multivariate logistic regression analyses were performed with SPSS Version 24 (IBM, Chicago, IL, USA).

## 3. Results

Response to a third vaccination after non-response to the standard vaccination schedule with two doses of a SARS-CoV-2 mRNA vaccine (BNT162b2 or mRNA-1273) was analyzed in 94 kidney transplant recipients. The median age was 63.1 years, and 64% of the patients were male. All included patients were Caucasian. The median eGFR was 41.6 mL/min/1.73 m^2^. Details on the immunosuppressive treatment and comorbidities are displayed in Table 1.

Table 2 shows the type of vaccine used and the results of the antibody measurement. Only one patient received a combination of mRNA-1273 and BNT162b2, while all other patients received the same mRNA vaccine three times (57% BNT162b2 and 42% mRNA-1273). The median time between the second and the third vaccination was 112 days, and the median time between the third vaccination and the antibody measurement was 28 days.

Non-response according to the definition of the manufacturer was 53%. In a univariate regression model including all parameters from Table 1, age, the type of vaccine (BNT162b2 vs. mRNA-1273), the time between the 2nd and the 3rd vaccination, the induction treatment in the peri-transplant phase (basiliximab vs. ATG), the type of organ donation (living vs. deceased graft), the number of previous kidney transplantations, the CMV and BK virus reactivation, double vs. triple immunosuppression, and the time since last kidney transplantation were not significantly associated with non-response; only cardiovascular disease was significantly associated with non-response (OR 1.86, *p* < 0.001). In the multivariate analysis, no significant associations were detected.

Figure 1 shows the high variation in measured antibody titers in the seroconversion group, with a median of 73.9 BAU/mL, IQR 221.5 (versus median 0.0, IQR 0.0 in the non-responding group).

## 4. Discussion

Since 2021, SARS-CoV-2 vaccination has become the standard of care for the prevention of severe COVID-19 disease and associated mortality. However, immunocompromised individuals show an impaired humoral immune response. In this study, we investigated the humoral immune response of kidney transplant recipients to a three-dose mRNA vaccination schedule, with the third dose offered as a timely booster in case of non-response after the completion of the standard two-dose vaccination regimen.

As national recommendations differ between countries, the timing of booster vaccination in kidney transplant recipients varies, limiting comparability. Similar to our analysis, where the third dose was administered after a median of 112 days after the second dose, several other studies analyzed the humoral immune response to a three-dose regimen with mRNA vaccines that included a timely booster dose after 28–74 days [17,18,20,23,24,25,26]. In a study from Israel, where single organ transplant recipients received a booster vaccine as the third dose at least five months after the second dose, 58% of the non-responders to the first two doses of mRNA vaccine seroconverted [27]. Recently, Yahav et al. published their results on a three-dose mRNA vaccination schedule in 190 kidney transplant recipients, with an overall seroconversion rate of 70% [28].

Regarding predictive factors for non-response in our data, only prevalent cardiovascular disease (CVD) was significantly associated with non-response in the univariate regression analysis. In the multivariate analysis, this association was no longer significant. Nevertheless, Naruse et al. already showed that patients with CVD presented an impaired humoral response after two doses of BNT162b2 [29]. The authors suggested a correlation with cardiologic medication, but the exact pathomechanism remained unclear. Masset et al. detected low lymphocyte count as a risk factor for non-response, whereas the use of antiproliferative drugs and steroids did not seem to significantly affect the seroconversion [24]. Accordingly, we did not find a statistically significant association between the type of immunosuppressive treatment and non-response. Nevertheless, significant correlation with seroconversion in kidney transplant recipients who discontinued treatment with an antimetabolite prior to a third vaccination had been observed [28,30,31]. Osmanodja et al. found that younger age, higher BMI, higher transplant age, higher eGFR, and higher hemoglobin levels were associated with improved serological response after a third vaccination [32].

Although approximately half of the kidney transplant recipients develop a humoral immune response after a third COVID-19 vaccination, the risk of severe COVID-19 disease still exists. A rapid decline in both cellular and humoral responses within six months was found in kidney transplant recipients with initial cellular and humoral responses one month after the third dose [33]. Recently, Kamar et al. published a retrospective analysis of antibody concentration up until three months after the third dose in a large cohort of SOT recipients: one month after dose 3, antibodies were detected in 66.3% of the patients. Of those, 94.5% remained seropositive after another three months, but binding and neutralizing antibody concentrations decreased significantly [34].

Our study has some limitations. The study was a non-randomized, retrospective analysis in a clinical routine setting and included only Caucasian patients. The development of cellular immunity was not assessed, although some evidence indicates that significant T-cell immunity in cases of humoral non-response might exist [35]. In our kidney transplant center, nearly all patients are on triple immunosuppression therapy including steroids. Therefore, our results might not be generalizable to organ transplant recipients on a steroid-free immunosuppression therapy.

In line with recent literature [17,18] we demonstrate that approximately half of the kidney transplant recipients without antibody response after a standard two-dose vaccination regimen showed humoral immune response after a third vaccination, with a high variation in detected antibody titers. Nevertheless, our results also indicate that a third vaccination does not induce sufficient serological response in a high percentage of kidney transplant recipients. Hence, strict sanitary protection measures (including hand hygiene and FFP2 masks) should be maintained and complete vaccination of household members is highly recommended for all patients, especially for non-responders to a schedule of three SARS-CoV-2 mRNA vaccines. These non-medical measures have been shown to reduce transmission rates [36]. Additionally, there are several therapeutic options, depending on the severity of disease and time after infection, including antiviral substances (remdesivir, nirmatrelvir/ritonavir, molnupiravir), immunomodulating drugs (baricitinib = JAK inhibitor, tocilicumab = monoclonal antibody against IL-6, corticosteroids), and neutralizing monoclonal antibodies. Moreover, there is a possibility to administer a pre-exposure prophylaxis with tixagevimab and cilgavimab to immunocompromised individuals [37].

While data on the fourth and fifth vaccinations in kidney transplant recipients are scarce, there is some evidence indicating that serological response might be improved through a repetitive vaccination regimen [32]. Further studies are clearly needed to identify adequate vaccination schedules in this highly vulnerable population.

## Figures and Tables

**Figure 1 jcm-11-06439-f001:**
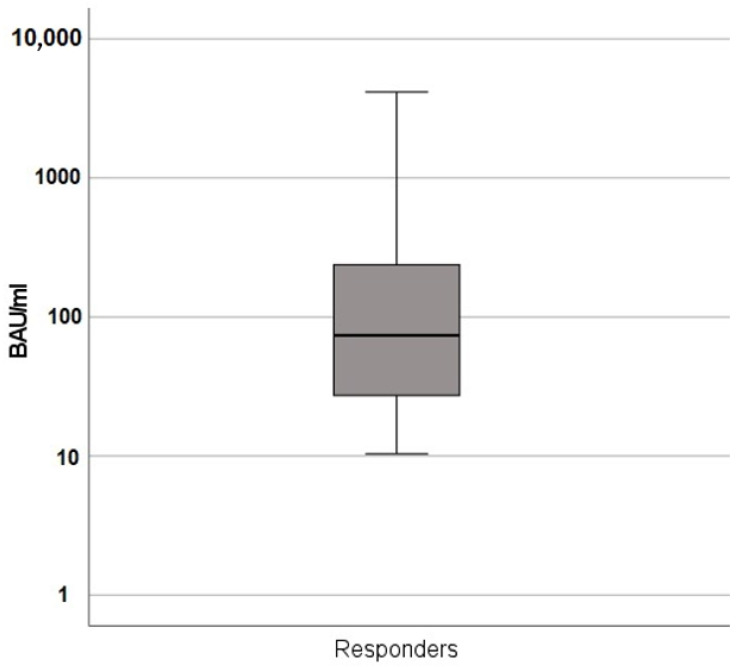
SARS-CoV-2 IgG titers in the seroconversion group (BAU/mL). Logarithmic scale.

**Table 1 jcm-11-06439-t001:** Baseline characteristics of 94 kidney transplant recipients. Continuous data are displayed as median and 25% and 75% percentiles. Numerical data are displayed as number of participants (*n*) and percentage (%) where appropriate (sums do not add up to 100% due to rounding).

Age (years)	63.1 (54.9–70.9)
Female (*n*, %)	34 (36)
BMI (kg/m^2^)	24.9 (22.0–28.8)
Comorbidities (*n*, %)	
Cardiovascular disease	40 (43)
Cerebrovascular disease	17 (18)
Active or former malignancy	21 (22)
Diabetes mellitus	40 (43)
Comedication (*n*, %)	
Treatment with RAAS inhibitors	42 (45)
High-dose glucocorticoid treatment during last year (≥1 mg/kg)	8 (9)
Tacrolimus	65 (69)
Cyclosporine A	17 (18)
Azathioprine	4 (4)
Mycophenolic acid	78 (83)
Belatacept	9 (10)
Glucocorticoids	88 (94)
mTor inhibitors	2 (2)
Laboratory values	
Albumin (g/dL)	4.1 (3.8–4.4)
Hemoglobin (g/dL)	131 (117–144)
C-reactive protein (mg/dL)	0.21 (0.09–0.39)
eGFR (mL/min/1.73 m^2^)	41.6 (30.6–60.3)

BMI: Body mass index; RAAS: Renin-angiotensin-aldosterone system inhibitors; eGFR: Estimated glomerular filtration rate.

**Table 2 jcm-11-06439-t002:** Vaccination strategy and antibody measurements. Continuous data are displayed as median and 25% and 75% percentiles. Numerical data are displayed as number of participants (*n*) and percentage (%) where appropriate (sums do not add up to 100% due to rounding).

Vaccine (*n*, %)	
3× BNT162b2	54 (57)
3× mRNA-1273	39 (42)
2× mRNA-1273 + 1× BNT162b2	1 (1)
Time between 2nd and 3rd vaccination (days)	112 (49–217)
Results of antibody measurement	
SARS-CoV-2 IgG titer (BAU/mL)	4.9 (0–4149.6)
Non-response (*n*, %)	50 (53)
Time between 3rd vaccination and AB measurement (days)	28 (28–30)

BNT162b2: Comirnaty, BioNTech/Pfizer vaccine; BAU: Binding antibody units; AB: Antibody.

## Data Availability

The data presented in this study are available on request from the corresponding author. The data are not publicly available due to data protection.

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
