# Peer review of "Humoral Immune Response to a Timely Booster mRNA Vaccination in Non-Responders to a Standard Vaccination Schedule against COVID-19 in Kidney Transplant Recipients"

_jcm, 2022, doi:10.3390/jcm11216439_

Round 1
Reviewer 1 Report
The authors present their experience with third dose of mRNA vaccine in kidney transplant recipients in Austria. They report about 50% seroconversion in non-responder transplant recipients after the third dose. This data has been reported by other groups and the outcomes reported in this study are similar to other reports and as such, not unique or new.
I do have a few questions/comments for the authors:
- Was the antibody test used to check antibody against the spike protein? This is not mentioned in the article.
- Did the authors look and compare the antibody response in recipients on triple immunosuppression Vs those on double immunosuppression? This has been reported by other groups.
- The authors do report about their univariate and multivariate analysis. I don't see that analysis reported here or the variables considered. Were age of the recipient or time after second dose used as a variable? It has been reported that transplant recipients not only have poor seroconversion but also have rapid attrition of antibody response over time. There isn't a lot of data about serial follow-ups of recipients who do turn seropositive, to see how rapidly their antibody levels decline. That would be a more interesting outcome to look at.
- Did the authors look at induction immunosuppression for the recipients who continued to be non-responders Vs those who seroconverted?
- Did the authors look at patients in the peri-transplant or early postoperative period? Did they have a poorer response than patients in late postoperative period due to higher maintenance immunosuppression and recent induction immunosuppression?
Overall, this study reports expected outcomes and is similar to other reported studies on this topic.
Reviewer 2 Report
This manuscript adds to the literature supporting vaccination amongst kidney transplant patients. Most of the patients from what I can tell are on steroid/three-drug immunosuppression, which is both a strength and weakness of the manuscript in the sense that while not entirely applicable to many transplant populations it does demonstrate that a response rate in patients on high dose immunosuppression. Other weaknesses to clinical translation are that the patient population is thin and I assume Caucasian, but it is not clear from the manuscript.
Round 2
Reviewer 1 Report
The authors have provided appropriate responses to the questions asked. Accept.